# I-PGD-AT: Efficient Adversarial Training via Imitating Iterative PGD Attack

## Abstract

To improve the efficiency of adversarial training, recent studies leverage Fast Gradient Sign Method with Random Start (FGSM-RS) for adversarial training. Unfortunately, such methods lead to relatively low robustness and *catastrophic overfitting*, which means that the robustness against iterative attacks (*e.g.* Projected Gradient Descent (PGD)) would suddenly drop to 0%. Different approaches have been proposed to address this problem, however, recent studies show that catastrophic overfitting still remains. In this paper, motivated by the fact that expensive iterative adversarial training methods achieve high robustness without catastrophic overfitting, we ask: *Can we perform iterative adversarial training in an efficient way?* To this end, we first analyze the differences in perturbations generated by FGSM-RS and PGD and find that PGD tends to craft *diverse* discrete values instead of $\pm 1$ in FGSM-RS. Based on this observation, we propose an efficient single-step adversarial training method I-PGD-AT by leveraging I-PGD attack, in which I-PGD imitates PGD virtually. Unlike FGSM that crafts the perturbations directly using the sign of gradient, I-PGD *imitates* the perturbation of PGD based on the magnitude of gradient. Extensive empirical evaluations on CIFAR-10 and Tiny ImageNet demonstrate that our I-PGD-AT can improve the robustness compared with the baselines and significantly delay catastrophic overfitting. Moreover, we explore and discuss the factors that affect catastrophic overfitting. Finally, to demonstrate the generality of I-PGD-AT, we integrate it into PGD adversarial training and show that it can even further improve the robustness.

## 1 Introduction

The breakthroughs in Deep Neural Networks (DNNs) have brought unprecedented boom to various fields, such as Computer Vision (Krizhevsky et al., 2012; He et al., 2016) and Natural Language Processing (Devlin et al., 2019). However, recent studies have revealed that DNNs are vulnerable to *adversarial examples*, such that imperceptible perturbations with small magnitude added to the input are sufficient to mislead the target model (Szegedy et al., 2014; Goodfellow et al., 2015; Xiao et al., 2018a;b; Bhattad et al., 2020). Such perturbations are often transferable across different models (Liu et al., 2017; Dong et al., 2018; Wang & He, 2021) and can be generated without the knowledge of the target model under black-box setting (Brendel et al., 2018; Li et al., 2020b; 2021).

Several works have been proposed to mitigate the aforementioned problem and improve the robustness of DNNs, *e.g.* adversarial training (Goodfellow et al., 2015; Madry et al., 2018), input transformations (Xie et al., 2018; Naseer et al., 2020), certified defenses (Raghunathan et al., 2018; Gowal et al., 2019) *etc*. Among these methods, adversarial training is one of the most effective empirical defense methods (Athalye et al., 2018). Adversarial training can be formulated as a minimax optimization problem (Madry et al., 2018), in which the inner maximization aims to search for an adversarial example that maximizes the classification loss, while the outer minimization aims to train a robust classifier against the worst-case adversarial perturbations. Projected Gradient Descent Adversarial Training (PGD-AT) (Madry et al., 2018) approximately solves the inner maximization problem by multiple steps of projected gradient descent and achieves empirical robustness (Carlini & Wagner, 2017; Athalye et al., 2018; Croce & Hein, 2020b). However, PGD-AT is much more expensive than standard training, owing to the iterative optimization process of perturbation generation.

To reduce the computational burden, Wong et al. (2020) propose Fast Adversarial Training (Fast-AT), which adopts Fast Gradient Sign Method with Random Start (FGSM-RS) for training and leads to

robust models. However, there are two limitations for Fast-AT: a) Fast-AT suffers from *catastrophic overfitting*, *i.e.* the robustness against iterative attack (*e.g.* PGD) suddenly drops after a few training epochs. b) There is still a gap between the robustness of Fast-AT and PGD-AT. Fast-AT directly adopts early stopping to prevent catastrophic overfitting. Several attempts (Andriushchenko & Flammarion, 2020; S. & Babu, 2020; Sriramanan et al., 2020; Kim et al., 2021) are made to avoid catastrophic overfitting, but these approaches either are computationally inefficient or hurt the robustness.

In this work, motivated by the benefits of PGD-AT, we propose I-PGD Adversarial Training (I-PGD-AT), to improve the robustness of Fast-AT with the same computational cost and delay catastrophic overfitting. In particular, we first analyze the differences in perturbations generated by FGSM-RS and PGD and find that the perturbation of PGD is more *diverse* in which the perturbations are different discrete values instead of $\pm 1$ in FGSM-RS. Based on this observation, we *imitate* the perturbation of PGD virtually via a single gradient calculation. Specifically, we first calculate all the candidate discrete values for PGD attack (*e.g.* 0 and $\pm 2$ for two-step PGD) and its corresponding probability. Then we generate a new gradient vector by filling the cells with the candidate discrete value based on the corresponding absolute gradient while keeping the original sign information. Unlike FGSM-RS which directly crafts adversarial perturbation using the sign of gradient, we leverage this generated vector with more *diverse* discrete values to generate adversarial perturbation for efficient training.

We conduct extensive empirical evaluations on CIFAR-10 and Tiny ImageNet datasets, which demonstrate that I-PGD-AT can significantly improve the robustness against various adversarial attacks. We also show that I-PGD-AT can effectively stabilize the training and delay catastrophic overfitting compared with Fast-AT. Furthermore, we empirically validate our hypothesis that *under the same $\ell_\infty$-norm perturbation constraint, training the model using perturbation with a smaller mean absolute value can delay catastrophic overfitting*, which could provide new insights to address such issues. Additionally, we extend our imitation strategy for multi-iteration, which achieves substantial improvements on the robustness compared with PGD-AT with different iterations, and delays the robust overfitting (Rice et al., 2020) in PGD-AT.

## 2 RELATED WORK

To mitigate the threat of adversarial examples, a large variety of defense method has been proposed, including adversarial training (Goodfellow et al., 2015; Madry et al., 2018), input transformations (Xie et al., 2018; Guo et al., 2018; Liu et al., 2019), denoising (Liao et al., 2018; Mustafa et al., 2019; Naseer et al., 2020), certified defense (Raghunathan et al., 2018; Cohen et al., 2019; Gowal et al., 2019; Li et al., 2019). Among them, PGD-AT (Madry et al., 2018), as the most popular adversarial training method, brings consistent improvements on robustness against various attacks and inspires several adversarial training methods (Zhang et al., 2019b; Ding et al., 2020; Wang et al., 2020; Song et al., 2020; Dong et al., 2020). Furthermore, Rice et al. (2020) identify *robust overfitting* in PGD-AT and find that early stopping or data augmentation (Tack et al., 2021) helps PGD-AT achieve comparable robustness with advanced adversarial training methods (Zhang et al., 2019b).

However, PGD-AT is computationally inefficient due to the iterative optimization for adversarial examples. To combat the computational overhead, recent works (Shafahi et al., 2019; Zhang et al., 2019a) utilize a single backpropagation for both training and PGD adversary generation to accelerate the training process. Wong et al. (2020) identify *catastrophic overfitting* in single-step adversarial training and propose Fast Adversarial Training (Fast-AT) for efficient training. To avoid catastrophic overfitting, Andriushchenko & Flammarion (2020) propose GradAlign, which maximizes the gradient alignment inside the perturbation set to strengthen the model's local linearity. However, it leads to $3\times$ slowdown due to the double backpropagation. Kim et al. (2021) identify decision boundary distortion after catastrophic overfitting, in which a smaller perturbation is sufficient to fool the model, while the model is robust against larger perturbation used in Fast-AT. They further propose to search appropriate step size for each input sample to avoid catastrophic overfitting, but it also hurts the robustness performance. Improved fast AT (FastAdv+) (Li et al., 2020a) incorporates PGD-AT into Fast-AT when observing the catastrophic overfitting to stabilize the training. Chen et al. (2020) propose *backward smoothing* which adopts a back-propogation for initialization to implement single-step training with the objtective of TRADES (Zhang et al., 2019b).

In this work, we identify the differences in perturbations generated by PGD and FGSM-RS and propose a *single-step* adversarial training method I-PGD-AT to boost the robustness of models and delay catastrophic overfitting effectively.

## 3 METHODOLOGY

In this section, we first provide preliminaries and revisit FGSM-RS and PGD to identify the differences in perturbations generated by them. Then we provide a detailed description of the proposed I-PGD Adversarial Training (I-PGD-AT) and a discussion on catastrophic overfitting.

### 3.1 PRELIMINARIES

Let $J(x, y, \theta)$ denote the loss function of deep neural net $f$ with parameters $\theta$ on the input $(x, y)$ drawn from data distribution $\mathcal{D}$. Adversarial training can be formulated as a minimax optimization problem (Madry et al., 2018):

$$\min_{\theta} \mathbb{E}_{(x,y) \sim \mathcal{D}}[\max_{\delta \in \Delta} J(x + \delta, y, \theta)], \tag{1}$$

where $\Delta$ is the perturbation set. In this paper, we focus on $\ell_{\infty}$ attack to align with previous work, *i.e.* $\Delta = \{\delta \in \mathbb{R}^d : \|\delta\|_{\infty} < \epsilon\}$, where $\epsilon$ is the maximum magnitude of the perturbation. This minimax problem is usually solved by firstly crafting adversarial examples to solve the inner maximization problem and then optimizing the model parameters $\theta$ using the generated adversarial examples.

Fast Gradient Sign Method Adversarial Training (FGSM-AT) (Goodfellow et al., 2015) is the first adversarial training method that solves the inner maximization problem using one-step update:

$$\delta = \epsilon \cdot \text{sign}(\nabla_x J(x, y, \theta)). \tag{2}$$

However, FGSM-AT exhibits limited robustness against iterative attacks (*e.g.* iterative FGSM (Kurakin et al., 2017)). Later, Madry et al. (2018) propose Project Gradient Descent Adversarial Training (PGD-AT), which iteratively solves the inner maximization problem using PGD:

$$\delta_{t+1} = \Pi_{\Delta}(\delta_t + \alpha \cdot \text{sign}(\nabla_x J(x + \delta_t, y, \theta))), \tag{3}$$

where $\delta_t$ is the adversarial perturbation at the $t$-th iteration and $\delta_0 \sim \mathcal{U}(-\epsilon, \epsilon)$, $\Pi_{\Delta}(\cdot)$ is the projection function and $\alpha < \epsilon$ is the step size.

Although PGD-AT is empirically robust, compared with FGSM-AT, the inner loop of PGD to craft adversarial examples is very expensive. Further, Wong et al. (2020) identify that FGSM-AT suffers from the *catastrophic overfitting* phenomenon, and propose Fast Adversarial Training (Fast-AT) using FGSM adversarial example with random start (FGSM-RS), *i.e.*, $\rho \sim \mathcal{U}(-\epsilon, \epsilon)$:

$$\delta = \Pi_{\Delta}(\rho + \alpha \cdot \text{sign}(\nabla_x J(x + \rho, y, \theta))). \tag{4}$$

Fast-AT adopts early stopping to avoid catastrophic overfitting.

### 3.2 REVISITING FGSM-RS AND PGD

Let $\delta' \in \mathbb{R}^d$ denote the perturbation obtained by gradient ascent and $g_i$ denote the gradient at $i$-th iteration calculated by the corresponding adversarial attack. Suppose that the input $x$ is normalized into $[0, 1]$, for existing gradient-based adversarial attacks, the generated adversarial example $x_{adv}$ can be formulated as:

$$x_{adv} = \Pi_{[0,1]}(x + \Pi_{\Delta}(\rho + \delta')), \tag{5}$$

where $\rho$ is the initialization of perturbation (*e.g.* $\rho = 0$ for FGSM and $\rho \sim \mathcal{U}(-\epsilon, \epsilon)$ for FGSM-RS and PGD). We could reformulate the final perturbation as $\delta = \Pi_{\Delta}(\rho + \delta')$.

For FGSM-RS, we could easily obtain $\delta'_{FGSM-RS} = \alpha \cdot \text{sign}(\nabla_x J(x + \rho, y; \theta)) = \alpha \cdot \text{sign}(g_0)$ from Eq. 4. For PGD$k$ where $k$ is the number of iteration of PGD, we can approximately obtain that:

$$\begin{aligned} \delta_k = \Pi_{\Delta}(\rho + \delta') &= \Pi_{\Delta}(\delta_{k-1} + \alpha \cdot \text{sign}(g_{k-1})) \\ &= \Pi_{\Delta}(\Pi_{\Delta}(\delta_{k-2} + \alpha \cdot \text{sign}(g_{k-2})) + \alpha \cdot \text{sign}(g_{k-1})) \\ &= \Pi_{\Delta}(\Pi_{\Delta}(\cdots \Pi_{\Delta}(\rho + \alpha \cdot \text{sign}(g_0))) + \alpha \cdot \text{sign}(g_{k-1})) \\ &\simeq \Pi_{\Delta}(\rho + \alpha \cdot \sum_{i=0}^{k-1} \text{sign}(g_i)), \end{aligned} \tag{6}$$

where $g_k = \nabla_x J(x + \delta_k, y, \theta)$. By ignoring the projection operator at each iteration as the step size $\alpha$ is expected to be small w.r.t. the input, and we can see that $\delta'_{PGDk} \simeq \alpha \cdot \sum_{i=0}^{k-1} \text{sign}(g_i)$ for PGD$k$.

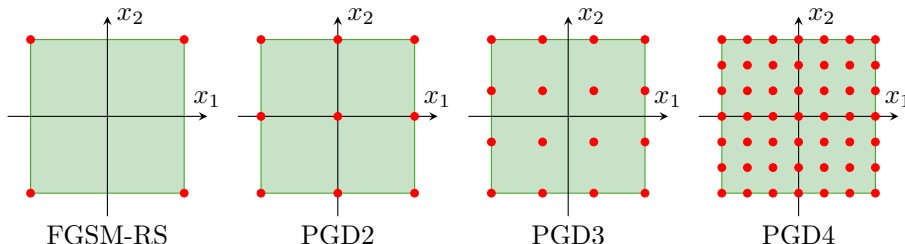

Figure 1: The candidate perturbation $\delta'$ (red dots) generated by gradient ascent in various attacks, namely FGSM-RS and PGD$k$, where $k$ denotes the number of iteration for PGD. For simplicity, we normalize the perturbation into $[-1, 1]$ and do not consider the hyper-parameter $\alpha$.

Since FGSM-RS and PGD adopt the same random initialization, we only focus on the difference of $\delta'$. For simplicity, we treat the step size $\alpha$ as a scaled factor and do not consider it by normalizing the accumulated sign $\delta'$ into $[-1, +1]$.

Suppose the input sample $x = (x_1, x_2) \in [0, 1]^2$, the perturbation set for $\delta'$ would be a square with side length 2 centered at the origin. Note that the output of $\text{sign}(\cdot)$ is either 1 or $-1$. As shown in Fig. 1, for FGSM-RS, $\delta'_{FGSM-RS}$ would be one of the four vertices of the square. In contrast, taking PGD2 for example, there are four cases $\langle +1, +1 \rangle, \langle +1, -1 \rangle, \langle -1, +1 \rangle, \langle -1, -1 \rangle$ of the sign for each pixel $x_i \in x$, which lead to three different *accumulated signs* $AS(2) = \{+2, 0, -2\}$. The general accumulated sign set for PGD$k$ can be formulated as:

$$AS(k) = \{\pm(2i + k\%2) : 0 \le i \le \lfloor k/2 \rfloor\}. \tag{7}$$

We show the search space for PGD2, PGD3 and PGD4 in Fig. 1 and we can see that the larger $k$ is, the more *diverse* the search space for PGD$k$ is. From the decision boundary distortion perspective (Kim et al., 2021), it supports that PGD$k$ can find more diverse adversarial examples inside the perturbation set so that it does not lead to catastrophic overfitting. Such difference of perturbation generated by FGSM-RS and PGD motivates us to consider the following question:

*Could we imitate the diverse discrete values of the perturbation generated by PGD$k$ with a single gradient calculation to improve the robustness of single-step adversarial training and mitigate the catastrophic overfitting?*

### 3.3 I-PGD ADVERSARIAL TRAINING

From Sec. 3.2, we find that FGSM-RS directly reaches the boundary of perturbation set, which is one possible reason for catastrophic overfitting (Kim et al., 2021) in Fast-AT that performs adversarial training with perturbation generated by FSGM-RS, while PGD can generate diverse perturbation inside the boundary. To some extent, adopting random initialization which leads to more diverse perturbation, can mitigate such phenomenon. Apart from catastrophic overfitting, the robustness is also weaker for Fast-AT compared with PGD-AT. Considering the properties of perturbation generated by PGD, we aim to explore how to improve the robustness of Fast-AT and mitigate catastrophic overfitting by imitating the perturbation of PGD.

It is noted that both FGSM-RS and PGD adopt the sign of the gradient without considering the relative magnitude of each element in the gradient vector $g \in \mathbb{R}^d$. Due to the multiple iterations, PGD leads to *diverse* but still discrete candidate perturbation for $\delta'$. In order to imitate the diverse values of PGD by single gradient calculation, we need to generate a new imitative gradient vector $\tilde{g} \in \mathbb{R}^d$ for crafting adversarial perturbation by additionally utilizing the magnitude of gradient. Generally speaking, to achieve this goal, there are two issues to address.

**What discrete values does $\tilde{g}$ contain?** To imitate a specific PGD$k$ attack, we can effectively calculate the accumulated sign set from Eq. 7. For instance, if $k = 2$, we have three values $\{-2, 0, +2\}$ which can be assigned to each element. For any $k$, the candidate values are symmetric regarding the origin. Thus, we only need to consider the absolute values for simplicity, *i.e.* we only have $\{0, 2\}$ for PGD2, and the assigned values keep the same sign with the corresponding gradient dimensions.

**Which value do we assign to the element $\tilde{g}[i]$?** After determining what values does $\tilde{g}$ contain, we can assign the value to each element to obtain $\tilde{g}$ by utilizing the relative magnitude of the gradient $g$.

---

**Algorithm 1** The general algorithm for I-PGD Adversarial Training

---

**Input:** Training data $\mathcal{D}$, loss function $J(x, y, \theta)$, training epochs $T$, magnitude of perturbation $\epsilon$, adversarial step size $\alpha$, learning rate $\eta$, imitative values and corresponding portions $\mathcal{I} = \{v_i : p_i\}$
1: Initialize model parameters $\theta$
2: **for** $t = 1 \rightarrow T$ **do**
3:     **for** each batch $(x, y) \subset \mathcal{D}$ **do**
4:         $\delta = \mathcal{U}(-\epsilon, \epsilon)$
5:         $g = \nabla_x J(x + \delta, y, \theta), \quad \tilde{g} = g$
6:         Calculate the quantile $Q$ on $|g|$ based on the portions $p_i \in \mathcal{I}$
7:         **for** $i$-th $q_i \in Q$ **do**                           $\triangleright$ $p_i$ elements in $|g|$ are smaller than $q_i$
8:            $\tilde{g}[q_{i-1} < |g| \leq q_i] = v_i$
9:         **end for**
10:         $\tilde{g} = \tilde{g} \cdot \text{sign}(g)$                                 $\triangleright$ $\tilde{g}$ share the same sign of $g$
11:         $\delta = \Pi_\Delta(\delta + \alpha \cdot \tilde{g})$                     $\triangleright$ $\delta = \Pi_\Delta(\delta + \alpha \cdot \text{sign}(g))$ for FGSM-RS
12:         $\theta = \theta - \eta \cdot \nabla_\theta J(x + \delta, y, \theta)$
13:     **end for**
14: **end for**

---

We only need to consider the absolute value. The smaller $|g|[i]$ is, the smaller value is assigned to $|\tilde{g}|[i]$. Specifically, we first calculate the probability of each possible value for PGD$k$. Let $p$ denote the probability that a specific element obtains the same sign (*i.e.* $\langle \cdots, +1, +1 \cdots \rangle$ or $\langle \cdots, -1, -1 \cdots \rangle$) in two consecutive iterations, and we assume that $p > 1/2$ due to the neighborhood constraint by adversarial attacks. We can calculate the probability for each value by summing the probability for each possible sequence of sign. For a given gradient $g$, we can calculate the quantile on $|g|$ according to the accumulated probability $P(|\delta'|[i] \leq v)$ for each candidate value $v$ and set all the elements in this quantile range with value $v$ to obtain $\tilde{g}$ for adversarial training. We denote adopting $\tilde{g}$ for attack as I-PGD attack and such adversarial training method as I-PGD Adversarial Training (I-PGD-AT). Taking PGD2 for example, we could get that the imitative value is $0$ and $2$ with the probability $P(|\delta'|[i] = 0) = 1 - p, P(|\delta'|[i] = 2) = p$, *i.e.* $\mathcal{I} = \{0 : 1 - p, 2 : 1\}$. We provide the detailed calculation for $\mathcal{I} = \{v_i : p_i\}$ in Appendix A.1 and overall algorithm of I-PGD-AT in Algorithm 1.

### 3.4 DISCUSSION ON CATASTROPHIC OVERFITTING

The major difference between FGSM-RS and I-PGD attack is that for each input sample, FGSM-RS generates the adversarial example only using the sign of gradient, while I-PGD adopts the imitative gradient vector $\tilde{g}$ of PGD to craft adversarial example. Since PGD-AT does not lead to catastrophic overfitting and our I-PGD-AT utilizes I-PGD for adversarial training, we wonder if such imitation could effectively avoid catastrophic overfitting.

Unfortunately, as discussed in Sec. 4.5, we find that catastrophic overfitting still occurs in I-PGD-AT. However, compared with Fast-AT, I-PGD-AT can stabilize the training process and delay catastrophic overfitting effectively. We further explore the relationship between the magnitude of perturbation and catastrophic overfitting and validate a new hypothesis that *under the same $\ell_\infty$-norm constraint on the perturbation, the perturbation with a smaller mean absolute value can effectively delay catastrophic overfitting.* This hypothesis sheds new light on how to avoid catastrophic overfitting in single-step adversarial training effectively.

## 4 EXPERIMENTS

### 4.1 EXPERIMENTAL SETTING

**Datasets and Models.** We conduct empirical evaluations on CIFAR-10 (Krizhevsky et al., 2009) and Tiny ImageNet[1] using PreAct ResNet-18 (He et al., 2016). All the images are normalized into $[0, 1]$.

**Baselines.** To verify the effectiveness of I-PGD-AT, we imitate PGD2 and PGD3 dubbed I-PGD2-AT and I-PGD3-AT with *single-step* adversarial training, respectively. We compare them with Free-AT (Shafahi et al., 2019) and four single-step adversarial training methods, *i.e.* Fast-AT (Wong et al., 2020), GradAlign (Andriushchenko & Flammarion, 2020), Kim *et al.* (Kim et al., 2021) and

---

[1]https://www.kaggle.com/c/tiny-imagenet

Table 1: Classification accuracy (%) and training time (min) of various single-step adversarial training methods and the proposed methods I-PGD2-AT and I-PGD3-AT on CIFAR-10 and Tiny ImageNet against white-box attacks ($\epsilon = 8/255$). We **bold** the highest classification accuracy and lowest training time which outperforms the runner-up by at least $0.2$. The proposed methods are in gray .

(a) Evaluations on CIFAR-10.

| Method | Clean | FGSM | PGD20 | PGD50 | MIM | CW20 | CW50 | AA | Time(min) |
|---|---|---|---|---|---|---|---|---|---|
| Free-AT | 78.75 | 71.89 | 45.57 | 45.34 | 45.85 | 43.60 | 43.51 | 41.55 | 120.8 |
| Fast-AT | 83.27 | 55.31 | 46.85 | 46.28 | 47.82 | 46.56 | 46.31 | 43.28 | **26.0** |
| GradAlign | 81.24 | 54.55 | 47.48 | 47.18 | 48.00 | 46.91 | 46.69 | 44.09 | 81.4 |
| Kim *et al.* | **88.91** | 51.60 | 37.71 | 37.25 | 38.79 | 39.22 | 38.90 | 36.07 | 29.2 |
| FastAdv+ | 83.17 | **76.68** | 47.10 | 46.69 | 47.48 | 46.40 | 46.22 | 43.50 | 230.1 |
| I-PGD2-AT | 81.41 | 55.31 | 48.26 | 47.96 | 49.32 | 47.19 | 46.95 | 44.47 | **26.2** |
| I-PGD3-AT | 81.42 | 55.39 | **48.82** | **48.34** | **49.82** | **47.61** | **47.25** | **44.70** | **26.2** |

(b) Evaluations on Tiny ImageNet.

| Method | Clean | FGSM | PGD20 | PGD50 | MIM | CW20 | CW50 | AA | Time(min) |
|---|---|---|---|---|---|---|---|---|---|
| Fast-AT | 46.80 | 21.18 | 17.82 | 17.56 | 17.88 | 16.16 | 16.10 | 14.11 | **114.9** |
| Kim *et al.* | **56.21** | 16.62 | 9.32 | 9.03 | 9.70 | 10.86 | 10.64 | 8.10 | 130.2 |
| I-PGD2-AT | 45.53 | **21.84** | **18.57** | **18.37** | **18.71** | **16.76** | 16.64 | 14.55 | **115.0** |
| I-PGD3-AT | 45.55 | **21.82** | **18.56** | **18.44** | **18.64** | **16.84** | **16.81** | **14.79** | **115.0** |

FastAdv+ (Li et al., 2020a). We also extend I-PGD into PGD2-AT and PGD7-AT to imitate more powerful iterative adversarial training and compare them with the corresponding PGD-AT methods.

**Training Details.** We use SGD optimizer with momentum of $0.9$ and decay factor of $5 \times 10^{-4}$, and adopt the perturbation budget $\epsilon = 8/255$ for adversarial training. For single-step adversarial training, we train the model with $30$ epochs on CIFAR-10 and $40$ epochs on Tiny ImageNet, using cyclic learning rate (Smith, 2017) with the maximum learning rate of $0.2$ and increasing until the half of the training epochs. For PGD$k$-AT, we train the model with $200$ epochs using the maximum learning rate of $0.2$, which decays to $0.02$ at $50$-th epoch. The probability $p$ is set to $2/3$ and $\alpha = \epsilon$ for I-PGD-AT.

**Evaluations.** We evaluate the robustness using various attacks, *i.e.* FGSM (Goodfellow et al., 2015), PGD (Madry et al., 2018), Momentum Iterative Method (MIM) (Dong et al., 2018), Carlini-Wagner (CW) (Carlini & Wagner, 2017) and AutoAttack (AA) (Croce & Hein, 2020b). More details about the experimental setting are summarized in Appendix A.3.

## 4.2 Evaluation on I-PGD-AT

To validate the effectiveness of the proposed I-PGD-AT, we empirically evaluate the robustness of I-PGD2-AT and I-PGD3-AT, which performs the *single-step* training to imitate PGD with different steps for perturbation optimization on CIFAR-10 and Tiny ImageNet.

**Results on CIFAR-10.** As Fast-AT may lead to catastrophic overfitting after $24$-th epoch on CIFAR-10 (Andriushchenko & Flammarion, 2020), we train Fast-AT multiple times and provide the best results. As shown in Table 1a, Kim *et al.* achieves the lowest robustness against various attacks but higher clean accuracy due to the smaller generated perturbation on average. Fast-AT exhibits better robustness than Free-AT while GradAlign and FastAdv+ achieve higher robustness than Fast-AT and do not lead to catastrophic overfitting. Our proposed I-PGD2-AT could stabilize the training process and achieve the highest robustness compared with the three baselines, without catastrophic overfitting during the training. It is noted that I-PGD3-AT, which only adopts different imitative values and corresponding portions (*i.e.* $\{1 : 5/9, 3 : 4/9\}$ for I-PGD3-AT and $\{0 : 1/3, 2 : 2/3\}$ for I-PGD2-AT), exhibits higher performance than I-PGD2-AT with the same computational cost. It further shows the reasonability and effectiveness of the proposed imitation strategy. As for the comparison of *training time*, although GradAlign improves the robustness, it leads to roughly $3\times$ slowdown compared with Fast-AT. Free-AT takes smaller computational cost for each epoch but larger training epochs, leading to longer training time. FastAdv+ takes much longer times due to the larger training epochs and using PGD10 to recover catastrophic overfitting. Our proposed methods take about the same training time as Fast-AT since we only process the calculated gradient with constant time complexity, and achieves the highest robustness.

Table 2: Classification accuracy (%) and training time (hour) of PGD-AT w/ and w/o the proposed I-PGD on CIFAR-10 against white-box attacks ($\epsilon = 8/255$) without early stopping. We **bold** the highest classification accuracy and lowest training time which outperforms the runner-up by at least 0.5 and mark the results of the proposed method in gray .

| Method | Clean | FGSM | PGD20 | PGD50 | CW20 | CW50 | MIM | AA | Time(h) |
|---|---|---|---|---|---|---|---|---|---|
| PGD2-AT | **84.78** | 50.28 | 39.31 | 38.64 | 40.78 | 39.58 | 39.22 | 36.52 | **4.3** |
| PGD4-AT | **84.83** | 50.32 | 39.91 | 39.42 | 41.29 | 42.20 | 39.90 | 37.12 | 7.0 |
| I-PGD2-AT$_{PGD2}$ | 82.84 | **52.13** | **44.06** | **43.59** | **45.09** | **43.14** | **42.97** | **40.55** | **4.4** |
| PGD7-AT | **82.27** | 53.17 | 45.89 | 45.41 | 46.88 | 44.74 | 44.55 | 42.23 | **11.1** |
| PGD14-AT | 82.02 | **53.78** | **47.07** | **46.76** | **48.01** | **45.47** | **45.29** | **43.01** | 20.5 |
| I-PGD2-AT$_{PGD7}$ | **82.32** | 53.17 | **46.67** | **46.28** | 47.38 | **45.26** | **44.98** | **42.91** | **11.3** |

**Results on Tiny ImageNet.** To further validate the effectiveness of I-PGD-AT, we perform similar experiments on Tiny ImageNet. Due to the high computational cost of GradAlign, here we do not take it as baseline. As shown in Table 1b, the results consistently demonstrate that our I-PGD-AT can improve the robustness over Fast-AT without extra computational cost. We find that the improvement of I-PGD3-AT over I-PGD2-AT is not so obvious as on CIFAR-10, which might due to the fact that the sign consistency on Tiny ImageNet is higher than that on CIFAR-10 as discussed in Sec. 4.4.

## 4.3 INTEGRATING I-PGD INTO PGD-AT

Though the primary goal of I-PGD-AT is to boost Fast-AT, the imitation strategy is not limited to *single-step* adversarial training and can be integrated with PGD-AT during each step. Specifically, at each iteration, we incorporate I-PGD2 into PGD2-AT and PGD7-AT, denoted as I-PGD2-AT$_{PGD2}$ and I-PGD2-AT$_{PGD7}$, to imitate more powerful iterative adversarial training, namely PGD4-AT and PGD14-AT. As shown in Table 2, I-PGD2-AT$_{PGD2}$ and I-PGD2-AT$_{PGD7}$ consistently achieve better robustness than the corresponding baselines PGD2-AT and PGD7-AT, with almost the same training time. Surprisingly, I-PGD2-AT$_{PGD2}$ exhibits much better robustness than PGD4-AT. I-PGD2-AT$_{PGD7}$ achieves slightly lower robustness than PGD14-AT, which takes double iterations for optimizing the inner maximization problem of adversarial training and doubles the training time.

## 4.4 SIGN CONSISTENCY FOR PGD

The proposed method I-PGD is based on the hypothesis that due to the neighborhood constraint by adversarial attacks, the probability $p$ that a specific element in the input obtains the same sign in two consecutive iterations would be high and stable. To validate this hypothesis, we define the sign consistency (SC) for i-th iteration of PGD as follows:

$$SC(i) = \frac{1}{N} \sum_{j=1}^{N} \mathbb{1}(\text{sign}(g_i[j]) = \text{sign}(g_{i+1}[j])),$$

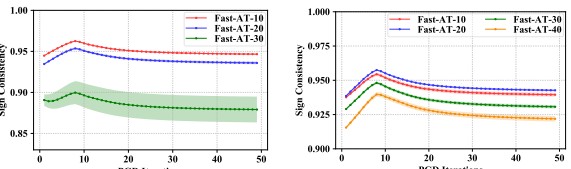

Figure 2: The sign consistency for PGD50 on Fast-AT at various epochs on CIFAR-10 (left) and Tiny ImageNet (right), where Fast-AT-$k$ denotes the model trained at $k$-th epoch. The results are averaged over 3 random seeds used for training and reported with the standard deviation. The high and stable sign consistency at various epochs makes it possible for imitation.

where $g_i$ denotes the gradient at $i$-th iteration and $\mathbb{1}(\cdot)$ is the indicator function. To some extent, $SC(i)$ also indicates the local linearity of the deep neural models. If $SC(i) = 1$, PGD attack would degenerate to FGSM-RS attack because the direction for perturbation update in Eq. 3 is constant at each iteration so that the iterations do not make any difference. In contrast, the perturbation of PGD differs from that of FGSM-RS significantly if $SC(i)$ is small. Besides, the stability of $SC(i)$ makes it possible for I-PGD-AT to adopt a fixed probability to imitate PGD-AT effectively.

We calculate $SC(i)$ for PGD50 attack on Fast-AT at various training epochs over 3 random seeds for training. As shown in Fig. 2, generally speaking, the sign consistency of PGD50 is high and stable at various iterations but training longer degrades the sign consistency and increases the standard deviation. Moreover, on Tiny ImageNet, the sign consistency decreases much slower with smaller standard deviation than that on CIFAR-10, which might be the reason why I-PGD3-AT cannot significantly enhance I-PGD2-AT and Fast-AT is more stable on Tiny ImageNet.

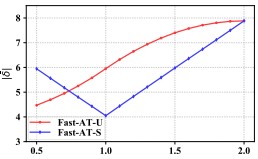

(a) The mean absolute value of perturbation $|\bar{\delta}|$.

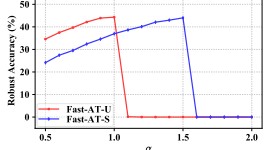

(b) Robust Accuracy against PGD50 attack.

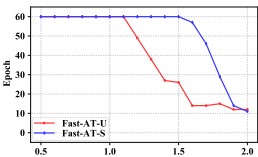

(c) The epoch when catastrophic overfitting occurs.

Figure 4: The (a) mean absolute value of perturbation $|\bar{\delta}|$ (normalized into $[0, \epsilon]$), (b) robust accuracy, and (c) the epoch when catastrophic overfitting occurs over various step size $\alpha$ for Fast-AT-U and Fast-AT-S. Initially, *catastrophic overfitting* does not happen and increasing $|\bar{\delta}|$ leads to better robustness. However, if *catastrophic overfitting* occurs, increasing $|\bar{\delta}|$ will boost *catastrophic overfitting*.

### 4.5 FACTORS RELATED TO CATASTROPHIC OVERFITTING

In this section, we investigate what factors would affect the catastrophic overfitting in single-step adversarial training. First, we explore if our I-PGD-AT can avoid catastrophic overfitting. We train the model with 60 epochs using two learning rate schedulers, *i.e.* single cyclic learning rate and the cyclic learning rate in the first 30 epochs followed with piecewise learning rate, in which the learning rate is decayed by a factor of $0.1$ per 10 epochs, dubbed CLR and CPLR respectively. As shown in Fig. 3, with CLR, catastrophic overfitting occurs

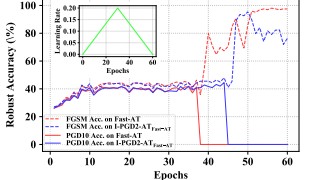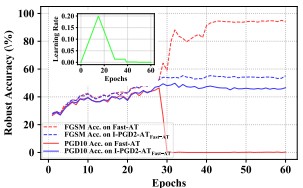

Figure 3: The robustness against FGSM (dashed line) and PGD10 (solid line) of Fast-AT and I-PGD2-AT over different training epochs with two different learning rate schedulers evaluated on randomly sampled 10% CIFAR-10 testset. The learning rate schedulers, namely Cyclic Learning Rate (CLR, left) and Cyclic Learning Rate followed by Piecewise Learning Rate (CPLR, right), are shown in the left upper corner in the corresponding figures. The red line and blue line denote the performance of Fast-AT and I-PGD2-AT trained over 60 epochs.

in both Fast-AT and I-PGD2-AT but I-PGD2-AT postpones it for 7 epochs compared with Fast-AT. Besides, with CPLR, Fast-AT still leads to catastrophic overfitting while I-PGD2-AT stably maintains its high robustness after 30th epochs. This indicates that our I-PGD-AT can effectively delay catastrophic overfitting and stablize the training process.

We further explore the factors that affect catastrophic overfitting on CIFAR-10. Previous works (Andriushchenko & Flammarion, 2020; Kim et al., 2021) show that small $\ell_\infty$-norm constraint $\epsilon$ for training prevents catastrophic overfitting. We find that even with the same $\epsilon$, the mean absolute value of the perturbation $|\bar{\delta}|$ generated by I-PGD2 is 6.5, which is a little smaller than the one of FGSM-RS. Furthermore, we find that adopting the sign of uniform perturbation for initialization also prevents catastrophic overfitting, which leads to $|\bar{\delta}| = 5.0$. Thus, we speculate that *under the same $\ell_\infty$-norm constraint on the perturbation $\delta$, smaller $|\bar{\delta}|$ can effectively delay catastrophic overfitting*.

To validate this hypothesis, we fix $\epsilon = 8/255$, while vary the step size $\alpha$ in Eq. 4 from $\epsilon/2$ to $2 \cdot \epsilon$ to adjust $|\bar{\delta}|$ for Fast-AT with *uniform perturbation* as initialization, dubbed Fast-AT-U. To show the generality of such hypothesis, we further adopt the *perturbation sampled from $\{-1, +1\}$ uniformly* as initialization, denoted as Fast-AT-S. As shown in Fig. 4a, when we increase the value of $\alpha$, $|\bar{\delta}|$ increases in Fast-AT-U while $|\bar{\delta}|$ first decreases before $\alpha = 1$, then increases linearly in Fast-AT-S. This is because when $\alpha = 1$, the random perturbation inconsistent with the sign of gradient will cancel each other out, resulting in the minimum perturbation. As shown in Fig. 4b, before the catastrophic overfitting occurring, the larger $\alpha$ which means larger $|\bar{\delta}|$ leads to better robustness for both initialization methods. If we continue to increase the value of $\alpha$, as depicted in Fig. 4c, the catastrophic overfitting occurs and the larger $\alpha$ leads to the earlier epoch for catastrophic overfitting. It supports the hypothesis and shows the trade-off between high robustness and catastrophic overfitting.

### 4.6 ABLATION STUDY

To further gain insights on the performance improvement obtained by I-PGD-AT, we conduct ablation study for the probability $p$ for I-PGD2-AT on CIFAR-10. As shown in Sec. A.1, the higher probability $p$ indicates the higher portion that the value 2 would occupy in $|\tilde{g}|$ at line 7-9 in Algorithm 1, leading

| p | 0.50 | 0.55 | 0.60 | 0.65 | 0.70 | 0.75 | 0.80 | 0.85 | 0.90 | 0.95 | 1.00 |
|---|------|------|------|------|------|------|------|------|------|------|------|
| Clean Acc. | 82.14 | 82.03 | 81.55 | 81.71 | 81.59 | 86.09 | 84.21 | 85.11 | 85.71 | 84.33 | 84.46 |
| PGD50 Acc. | 46.18 | 46.99 | 47.40 | 48.09 | 48.16 | 0.00 | 0.00 | 0.00 | 0.00 | 0.00 | 0.00 |
| $|\bar{\delta}|$ | 5.90 | 6.12 | 6.31 | 6.50 | 6.70 | 6.90 | 7.10 | 7.29 | 7.48 | 7.68 | 7.88 |
| Epoch | – | – | – | – | – | 28 | 23 | 20 | 14 | 10 | 10 |

Table 3: Ablation study for the probability $p$ used in I-PGD2-AT on CIFAR-10. $|\bar{\delta}\|$ denotes the mean absolute value of perturbation and Epoch denotes the epoch when catastrophic overfitting occurs

to larger $|\bar{\delta}|$. We vary $p$ from 0.5 to 1 and summarize the results in Table 3. As we can see, when we increase the value of $p$, the clean accuracy decreases a little but the robustness increases significantly when $p \leq 0.7$. However, when we continue to increase $p$, $|\bar{\delta}|$ increases and the catastrophic overfitting happens. The larger $p$ indicates the larger $|\bar{\delta}|$ and results in the earlier epoch for catastrophic overfitting. This is also consistent with our experiments and analysis in Sec. 4.5 and further support our hypothesis that *under the same $\ell$-norm constraint on the perturbation $\delta$, smaller $|\bar{\delta}|$ can effectively delay the catastrophic overfitting*.

### 4.7 ROBUST OVERFITTING IN I-PGD-AT

Rice et al. (2020) identify *robust overfitting* in PGD-AT that the robust accuracy begins to decrease as training progresses after a few epochs. Here we briefly explore whether I-PGD-AT is helpful to delay or avoid the robust overfitting. As shown in Fig. 5, I-PGD2-AT consistently exhibits better robust accuracy than PGD2-AT and PGD4-AT throughout the training process. As for the robust overfitting, I-PGD-AT maintains the peak performance between 50-th and 75-th epochs, while the robust accuracy of PGD2-AT and PGD4-AT begins to decrease around 50-th epoch. Hence, it is clear that I-PGD-AT can delay (not avoid) the robust overfitting in PGD-AT with better final robustness, and the sustainability for peak robustness might be helpful to choose a better epoch for early stopping.

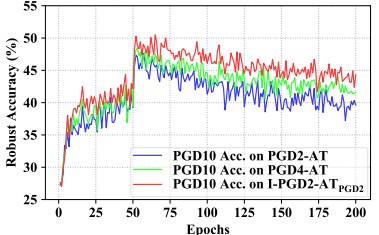

Figure 5: The robust accuracy (%) against PGD10 attack over the training epochs for PGD2-AT, PGD4-AT and I-PGD2-AT$_{PGD2}$.

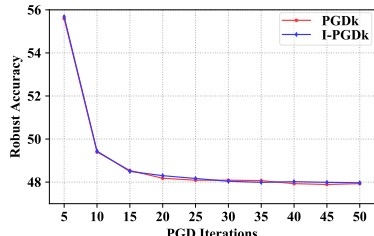

Figure 6: The robust accuracy (%) of I-PGD2-AT on CIFAR-10 against PGDk and I-PGDk attacks where $k$ denotes the number of iteration.

### 4.8 COMPARING I-PGD WITH PGD ATTACKS

Though we have observed that I-PGD-AT can significantly improve the robustness of single-step and multi-step adversarial training, we wonder whether I-PGD attack indeed imitates the attack effectiveness of PGD. To verify this, we incorporate the imitation strategy for PGD2 into PGD attack to evaluate the robustness of I-PGD2-AT on CIFAR-10. We denote such attack as I-PGD$k$ where $k$ is the number of iteration. As illustrated in Fig. 6, we find that for various number of iteration $k$, I-PGD$k$ consistently achieves similar attack performance as PGD$k$. The imitation for PGD3 also exhibits the same trend and we do not show it in the figure for clarity. The reasons should be two-fold: a) As shown in Fig. 6, increasing the number of iteration cannot effectively improve the attack performance for PGD$k$ when $k \geq 20$. b) Our imitation only considers the diverse value of perturbation without effectively predicting the sign of generated perturbation at each iteration, which might limit the attack performance. We believe that it is possible to improve the attack performance through imitating powerful attacks and leave this as our future work.

## 5 CONCLUSION

In this work, we identified the differences in perturbations crafted by FGSM-RS and PGD. Based on this observation, we proposed I-PGD adversarial training (I-PGD-AT) by adopting I-PGD attack that imitates PGD virtually through single gradient calculation. Experiments on CIFAR-10 and Tiny ImageNet showed that I-PGD-AT could effectively improve the robustness and delay *catastrophic overfitting* and robust overfitting. We believe that this work would inspire more precise imitation of PGD, which can improve the attack effectiveness and further enhance the robustness of DNNs.

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

## A    APPENDIX

### A.1    CALCULATION FOR $\mathcal{I} = \{v_i : p_i\}$

In this section, we first provide the detailed calculation for $\mathcal{I} = \{v_i : p_i\}$ with PGD2 as an example and then summarize a general algorithm for PGD$k$.

Taking PGD2 for example, there would be four cases of the sign for each pixel $x_i \in x$, namely $\langle +1, +1 \rangle, \langle +1, -1 \rangle, \langle -1 + 1 \rangle$ and $\langle -1, -1 \rangle$. Suppose the sign of the first iteration is either $+1$ or $-1$ with equal probability, we could calculate the probability for each sign sequence as following:

$$P(\langle +1, +1 \rangle) = \frac{1}{2} \cdot p = \frac{p}{2}, \qquad P(\langle +1, -1 \rangle) = \frac{1}{2} \cdot (1 - p) = \frac{1 - p}{2},$$

$$P(\langle -1, -1 \rangle) = \frac{1}{2} \cdot p = \frac{p}{2}, \qquad P(\langle -1, +1 \rangle) = \frac{1}{2} \cdot (1 - p) = \frac{1 - p}{2},$$

where $p$ is the probability that a specific element $x_i$ obtains the same sign (*i.e.* $\langle \cdots + 1, +1 \cdots \rangle$ or $\langle \cdots - 1, -1 \cdots \rangle$) in two consecutive iterations. The imitative value would be $\{0, 2\}$ and we could calculate the corresponding probability:

$$P(v = 0) = P(\langle +1, -1 \rangle) + P(\langle -1, +1 \rangle) = 1 - p$$
$$P(v = 2) = P(\langle +1, +1 \rangle) + P(\langle -1, -1 \rangle) = p$$

Thus, we could obtain that $\mathcal{I}[0] = P(v \le 0) = 1 - p$ and $\mathcal{I}[2] = P(v \le 2) = 1 - p + p = p$, *i.e.* $\mathcal{I} = \{0 : 1 - p, 2 : 1\}$. For an arbitrary value of $k$, we first traverse each possible sign sequence to calculate the corresponding probability. Then we calculate the probability for each potential imitative value by accumulating the probability of related sign sequences. To obtain the quantile for each possible value, we further accumulate the probability based on the imitative value in ascending order. The general calculation for any $k$ is summarized in Algorithm 2.

---

**Algorithm 2** Calculation for imitative value and accumulated probability: $\mathcal{I} = \{v_i : p_i\}$

---

**Input:** The number of iteration $k$ for imitative PGD, the probability $p$ that a specific element obtains the same sign in two consecutive iterations
1: Initialize $\mathcal{I} = \{\}$
2: **for** $i$-th binary sign sequence $s_i$ with value $\{+1, -1\}$ and size of $k$ **do**
3:      $v_i$ = the absolute value of the sum of $s_i$
4:      $p_{v_i}$ = the probability for the sign sequence $s_i$
5:      **if** $v \in \mathcal{I}$ **then**
6:          $\mathcal{I}[v_i] = 0$                            ▷ Initialize the probability for value $v_i$ in $\mathcal{I}$
7:      **end if**
8:      $\mathcal{I}[v_i] = \mathcal{I}[v_i] + p_{v_i}$               ▷ $v_i$ might be related to various sign sequences
9:      Sort $\mathcal{I}$ based on the key $v$ in ascending order
10:      **for** each $(v_i, p_i) \in \mathcal{I}$ **do**             ▷ Accumulate the probability as the quantile
11:          $\mathcal{I}[v_i] = \sum_{j=1}^{i} p_j$
12:      **end for**
13:      **return** $\mathcal{I} = \{v_i : p_i\}$
14: **end for**

---

### A.2    AN EXAMPLE OF CALCULATING THE QUANTILE

To better illustrate the algorithm, we also provide a simple example of imitating PGD2 with the probability $p = 2/3$ as follows.

First, we could obtain the probability $P(|\delta'|[i] = 0) = 1 - p = 1/3$, $P(|\delta'|[i] = 2) = p = 2/3$ and calculate the imitative value and accumulated probability $\mathcal{I} = \{0 : 1/3, 2 : 1\}$. Suppose we obtain the gradient matrix $g = \begin{bmatrix} 0.2 & -0.3 & -0.1 \\ 0.5 & -0.6 & 0.3 \\ 0.7 & 0.1 & 0.9 \end{bmatrix}$ at line 5 of Algorithm 1, we need to calculate the

start and end value for the range with the portion from 0 to $1/3$ and from $1/3$ to 1 on the sorted array $[0.1, 0.1, 0.2, 0.3, 0.3, 0.5, 0.6, 0.7, 0.9]$ (the absolute values of $g$), respectively. Thus, we could get the quantile $Q = \{0.1, 0.2, 0.9\}$ at Line 6 of Algorithm 1 (0 is the minimal value for the portion of 0, 0.2 is the smaller 3-quantile for the portion of $1/3$ and 0.9 is the maximum value for the portion of 1.). After the for-loop at line 7-9 of Algorithm 1, we have $\tilde{g} = \begin{bmatrix} 0 & 2 & 0 \\ 2 & 2 & 2 \\ 2 & 0 & 2 \end{bmatrix}$ by calculating each element $\tilde{g}[i] = \begin{cases} 0 & \text{if } 0.1 \leq |g|[i] \leq 0.2 \\ 2 & \text{if } 0.2 < |g|[i] \leq 0.9 \end{cases}$. Then we let $\tilde{g}$ share the same sign of $g$ at line 10 and obtain $\tilde{g} = \begin{bmatrix} 0 & -2 & 0 \\ 2 & -2 & 2 \\ 2 & 0 & 2 \end{bmatrix}$, which could be used to generate the perturbation for adversarial training at line 11-12 of Algorithm 1.

### A.3 EXPERIMENTAL SETTING

In this section, we provide more details about the experimental setting, including datasets, hyper-parameters for baselines and the attack settings.

**Datasets.** We adopt two widely investigated benchmark datasets for the evaluation, namely CIFAR-10 and Tiny ImageNet. CIFAR-10 contains 10 classes with $5,000$ training and $1,000$ testing images of resolution $32 \times 32$ per class. Tiny ImageNet contains $120,000$ images of 200 classes downsized to $64 \times 64$ colored images, in which each class has 500 training images, 50 validation images and 50 testing images.

**Hyper-parameters for baselines.** For Free-AT, we set $\alpha = \epsilon$, $m = 8$ and train the model for 200 epochs. We adopt $\alpha = 1.25 \cdot \epsilon$ for Fast-AT (Wong et al., 2020), set the regularization parameter for GradAlign to $0.2$ (Andriushchenko & Flammarion, 2020) and utilize 3 checkpoints for Kim *et al*. (Kim et al., 2021). We follow the setting of FastAdv+ (Li et al., 2020a) which reports the best results among 200 epochs with piecewise learning rate and adopts PGD-10 to recover the catastrophic overfitting. For PGD$k$-AT, the step size $\alpha$ is set to $\max(2/255, \epsilon/k)$. For all the baselines, we evaluate the effectiveness of the models at the last training epoch (Wong et al., 2020; Andriushchenko & Flammarion, 2020; Kim et al., 2021).

**Attack settings.** CW is implemented by adopting the margin-based loss function (Carlini & Wagner, 2017) and using PGD for optimization. We use 20 and 50 steps for PGD and CW and 50 steps for MIM with 10 random restarts. The perturbation budget $\epsilon$ is set to $8/255$ and step size $\alpha$ is set to $2/255$. AutoAttack adopts the default setting (Croce & Hein, 2020b) with four adversarial attacks, namely untargeted APGD-CE, targeted APGD-DLR, targeted FAB (Croce & Hein, 2020a) and Squared attack (Andriushchenko et al., 2020).

