# OpenReview forum: "I-PGD-AT: Efficient Adversarial Training via Imitating Iterative PGD Attack "
_ICLR.cc/2022/Conference — ICLR 2022 Submitted_

### Official Review · Reviewer_hVJW · 2021-10-29

**Correctness:** 3
**Technical Novelty And Significance:** 4
**Empirical Novelty And Significance:** 4
**Recommendation:** 6
**Confidence:** 4

**Main Review:**

**Strengths**:
* Great empirical results by improving over Fast-AT with the same runtime as well as delaying catastrophic overfitting.
* The paper is well written.
* The idea of using the quantiles of the gradient magnitudes to produce multiple step sizes  within a single step adversarial method is a clever idea.
* The code is attached to the supplementary material and looks usable for easy reproducibility.

**Weaknesses**:
1) Regarding equation (6) and "We approximately ignore the projection operator at each iteration as the step size $\alpha$ is expected to be small", the theoretical justification does not seem well-supported. First, according to the experiment section, $\alpha$ is not small enough compared to $\epsilon$. Indeed, if the initial random perturbation $\rho$ is not equal to 0, the perturbation can get "stuck" on the pertubation set boundary after a few steps. If $\rho$ is close to $\epsilon$, the perturbation boundary can even be reached after one step. Hence, we cannot ignore the projections. Second, if we cannot ignore the projections, we lose the grid structure (like in Figure 1) used in the rest of the analysis to determine the algorithm. I might be wrong for both points but it would require some clarifications.
2) Regarding the sign consistency experiment. Could the high value for the sign consistency be due to the perturbation being "stuck" on the perturbation set boundary? If yes, the fact that the projections which were ignored in order to come up with the algorithm would be the responsible for the high value of the sign consistency. That would go against "The proposed method I-PGD is based on the hypothesis that due to the neighborhood constraint by adversarial attacks, the probability $p$ that a specific element in the input obtains the same sign in two consecutive iterations would be high and stable".
3) The theoretical approach boils down to a simple result "{0 : 1 − p, 2 : p}" for  I-PGD2-AT and  "{1 : 1 − p, 3 : p}" for I-PGD3-AT. There might be other approaches leading to this simple result without making the assumptions discussed above which need clarifications. It could just be an heuristic approach such as: PGD leads to perturbations close to the boundary or closer to the random start. This last sentence translates into your result "{0 : 1 − p, 2 : p}" and would just require a sweep over the probability $p$. Such a sweep is done in Table 3 for I-PGD2-AT and it would be great to have the same for "{1 : 1 − p, 3 : p}" with I-PGD3-AT.
4) The ablation study on the probability $p$ in the appendix should have been in the main paper, more than the catastrophic overfitting discussion.
5) Minor. There are a few typos in the paper. Page 5 and in the appendix: {0 : 1 − p, 2 : 1} ->  {0 : 1 − p, 2 : p}. In Section 4.6, the subscripts of  "I-PGD2-AT" are lacking compared to Figure 5.
6) Minor. Why doing the catastrophic overfitting study in Figure 4 on Fast-AT rather than on the proposed method?
7) Minor. In Figure 5, you should add a curve "PGD10 Acc on PGD10-AT" to the plot for a more complete comparison.

**Summary Of The Paper:**

**Few sentences summary**: the paper proposes a single-step adversarial training method called I-PGD-AT to improve over the Fast Adversarial Training method. Based on the fact that not all the perturbations lie on the boundary of the perturbation set when using multiple steps PGD, this method aims at replicating this behaviour while still using only one single adversarial training step.

For the **proposed method**, it uses:
* Quantiles over the magnitudes of the gradient across the image.
* A model approximating how the perturbation space is "crowded" after $k$ PGD steps under the assumption that gradients sign changes between iterations occur with a fixed probability $p$.
* Quantiles over the gradient magnitudes and quantiles over the occupation of the perturbation space are matched such that the strongest gradients are assigned to the furthest position in the perturbation space.

For the **experiments**:
* Evaluation on CIFAR-10 and TinyImageNet against other single-step methods: Fast-AT, GradAlign and Kim et al.(2021).
* Evaluation on the extension of the proposed method to the PGD case.
* The paper evaluates the sign consistency of the gradients between two consecutive iterations as the proposed method relies on the assumption that sign changes between iterations occur with a fixed probability $p$.
* General investigation on catastrophic overfitting for single step methods.


**Summary Of The Review:**

The paper proposes a method which could be very useful to the community as it provides a single-step adversarial training method with clearly improved performance compared to existing methods like Fast-AT with the same runtime. I am vouching for acceptance due to the great experimental results but currently I am not yet satisfied by the theoretical justification of the method as a few statements are not well-supported. I think that it can be clarified in the rebuttal as using the quantiles of the gradient magnitudes to produce multiple step sizes  within a single step adversarial method is a clever idea.

---

> ### Author Response · Authors · 2021-11-23
> **Response to Reviewer hVJW (1/2)**
>
> > Q1: Regarding equation (6) and "We approximately ignore the projection operator at each iteration as the step size $\alpha$ is expected to be small," the theoretical justification does not seem well-supported.
>
> Our motivation and novelty are to imitate the perturbation generated by iterative PGD attack with single gradient calculation to improve the effectiveness of adversarial training. Thus, in Eq. (6), we recursively expand the equation for PGD attack and approximately omit the projection operator at each iteration to get the final sum, which is rough but motivates us to analyze and utilize the sign sequence to imitate the iterative PGD. The primary purpose of Eq. (6) is to describe our motivation that we could imitate the perturbation generated by iterative PGD with a single gradient calculation rather than to drive a precise approximation of PGD attack. And the empirical evaluation validates the effectiveness of our method.
>
> + According to the experiment section, $\alpha$ is not small enough compared to $\epsilon$.
>
>     $\alpha=\epsilon/4$ is generally accepted for PGD-AT, which is small w.r.t. the input, since $\epsilon$ should be small compared with the input. We have clarified it in the revision and would show the relative magnitude of $\alpha$ to $\epsilon$ is not crucial in the following.
>
> + Indeed, if the initial random perturbation $\rho$ is not equal to 0, the perturbation can get "stuck" on the perturbation set boundary after a few steps. If $\rho$ is close to $\epsilon$, the perturbation boundary can even be reached after one step. Hence, we cannot ignore the projections.
>
>     It is obvious that the projection does not make any difference for the perturbation within the boundary. **Thus, we could ignore the projection if the perturbation is in the boundary**. The main difference is that the perturbation could get stuck on the perturbation set boundary with projection as you said, while it would go far away from the boundary without the projection. Without loss of generality, let $\delta_t^{wp}$ and $\delta_{t+1}^{wp}$ be the perturbations without projection while $\delta_{t}^{p}$ and $\delta_{t+1}^{p}$ be the perturbations with projection in two consecutive iterations, in which the attack reach the boundary at t-th iteration, i.e., $\delta_t^{wp}=\delta_{t}^{p}$, $|\delta_{t+1}^{wp}|>\epsilon$ and $|\delta_{t+1}^{p}| = \epsilon$. Note that there are only two directions (+1, -1) since PGD adopts the sign of the gradient. There would be two cases:
>
>     + *The perturbation gets stuck on the perturbation boundary*. If the signs of the gradient of the data points for $\delta_t^{wp}$ and $\delta_{t+1}^{wp}$ are same, $\delta$ would always be outside the boundary with the same direction (-1 or +1) as $\delta_t^{wp}$ and would obtain the same data point on the boundary after the final projection. Otherwise, it would be get stuck between $\delta_t^{wp}$ and $\delta_{t+1}^{wp}$ and lead to slightly different data points if we finally obtain $\delta_t^{wp}$, which is close to the data point on the boundary because $\alpha$ is small.
>
>     + *The perturbation does not get stuck on the boundary*. In this case, the sign of the gradient for the data point $\delta_{t}^{p}$ is different from $\delta_{t+1}^{p}$. Since $\alpha$ is small, the sign of the gradient for $\delta_{t}^{p}$ would be consistent to the one for $\delta_{t+2}^{p}$ with high probability, i.e., the perturbation would get stuck between $\delta_{t}^{p}$ and $\delta_{t+2}^{p}$. Without projection, the perturbation after the final projection would be either $\delta_{t+1}^{p}$ (always outside the boundary) or $\delta_{t}^{p}$ (stuck between $\delta_t^{wp}$ and $\delta_{t+1}^{wp}$). Thus, the maximum difference between the two final perturbations would be $\alpha$, which is very small w.r.t. the input. Also, it is possible that with the projection, the perturbation changes its direction and continually moves towards the opposite direction while the perturbation without projection gets stuck between $\delta_t^{wp}$ and $\delta_{t+1}^{wp}$ or stays outside the boundary. This would be responsible for the main approximation error but happens with a very low probability.
>
>     Thus, we could ignore the projection in Eq. (6) for approximation.
>
> + Second, if we cannot ignore the projections, we lose the grid structure (like in Figure 1) used in the rest of the analysis to determine the algorithm.
>
>     Based on the above analysis, we could approximately ignore the projection of PGD at each iteration to obtain the grid structure in Fig. 1. The effectiveness of the I-PGD-AT also supports the rationality of the analysis.

---

> > ### Author Response · Authors · 2021-11-23
> > **Response to Reviewer hVJW (2/2)**
> >
> > > Q2: Could the high value for the sign consistency be due to the perturbation being "stuck" on the perturbation set boundary? If yes, the fact that the projections which were ignored in order to come up with the algorithm would be the responsible for the high value of the sign consistency.
> >
> > The sign consistency is based on the widely accepted hypothesis about the local linearity and smoothness of the DNNs. As shown in Fig. 2, the sign consistency is high and stable at the beginning of PGD attack, which does not reach the boundary. We also try to sample two random data points with the distance of $\alpha$ in the neighborhood of input on CIFAR-10, which also shows the high sign consistency. Thus, the high value of the sign consistency is not due to the perturbation being "stuck" on the perturbation set boundary, which also supports the rationality of our hypothesis.
> >
> > > Q3: The theoretical approach boils down to a simple result "{0 : 1 − p, 2 : p}" for I-PGD2-AT and "{1 : 1 − p, 3 : p}" for I-PGD3-AT.
> >
> > Based on the theoretical analysis, we propose a simple yet effective imitation method to improve Fast-AT. As you said, there might be other possible explanation that leads to similar results for imitation. However, what you assume is based on our results, and it might be hard to determine the candidate value (0 and 2 for I-PGD2-AT, 1 and 3 for I-PGD3-AT) without the analysis. Also, the performance improvement of I-PGD3-AT on I-PGD2-AT with different candidate values and probabilities supports the theoretical analysis, which could not be explained by your heuristic approach.
> >
> > > Q4: The ablation study on the probability $p$ in the appendix should have been in the main paper, more than the catastrophic overfitting discussion.
> >
> > We have reordered the experimental section and put the ablation study on the probability $p$ in the main paper. Thanks for your valuable suggestion.
> >
> > > Q5: Few typos.
> >
> > We have added the subscript of "I-PGD2-AT" in Sec. 4.6 and polished the entire paper again. $\mathcal{I}$ records the accumulated probability for calculating the quantile instead of the probability for each candidate value. We have clarified it in the appendix, and $\mathcal{I}=\\{0: 1-p, 2: 1\\}$ is correct in the paper. Thanks very much.
> >
> > > Q6: Why doing the catastrophic overfitting study in Figure 4 on Fast-AT rather than on the proposed method?
> >
> > As shown in Fig. 3, we find that the proposed I-PGD-AT could effectively delay the catastrophic overfitting in Fast-AT. Considering the difference between the perturbation generated by I-PGD and FGSM-RS, we speculate that *under the same $\ell_\infty$-norm perturbation constraint, training the model using perturbation with a smaller mean absolute value can delay catastrophic overfitting in Fast-AT*. To validate this hypothesis, we adjust the value of $\alpha$ and adopt two different initialization methods to obtain the perturbation with various mean absolute values on Fast-AT. We aim to explore a possible reason for the catastrophic overfitting in Fast-AT, which might shed light on avoiding catastrophic overfitting or improving Fast-AT for future works.
> >
> > > Q7: In Figure 5, you should add a curve "PGD10 Acc on PGD10-AT" to the plot for a more complete comparison.
> >
> > We plot the curve "PGD10 Acc on PGD2-AT" and "PGD10 Acc on PGD4-AT" in Fig. 5 because our I-PGD2-AT$_{PGD2}$ takes the same computational cost as PGD2-AT while imitating PGD4-AT. We only adopt PGD10 attack to evaluate the robust accuracy at each epoch because it takes less iterations compared with PGD20 or PGD50 used in Table 1. It seems to be not necessary to plot the curve "PGD10 Acc on PGD10-AT".

---

### Official Review · Reviewer_AAHj · 2021-10-31

**Correctness:** 3
**Technical Novelty And Significance:** 4
**Empirical Novelty And Significance:** Not applicable
**Recommendation:** 6
**Confidence:** 4

**Main Review:**

##########################################################################

Pros:

1. The problem the paper is trying to solve is an important one. Adversarial robustness is of great importance and adversarial training has
been shown to be one of the most effective methods. However, the efficiency of multi-step adversarial training is a problem.


2. The intuition behind the proposed method makes sense. The difference between PGD generated adversarial examples and FGSM generated ones comes from the perturbations. The proposed method introduces a heuristic way to imitate PGD adversarial perturbation based on one gradient query, which is efficient and novel.


3. Experimental results on CIFAR10 and TinyImageNet show that I-PGD-AT outperforms several fast adversarial training methods against state-of-the-art attacks. Besides, it does increase the training time much compared to FAT. Studies on catastrophic overfitting are also done to investigate contributing factors to the issue.


##########################################################################

Cons:


1. My main concern is that the method is not evaluated on ImageNet. In the paper of Fast adversarial training (Wong et al., 2020), the method was tested on ImageNet. It might be hard to address this problem as performing adversarial training (even the fast version) may take several days.


2. Overall, the paper is clearly written and well-organized, but some details are missing, making it hard to follow. (See detailed comments below.) But, this can addressed during rebuttal.

##########################################################################

Detailed Comments:

(1) I like the fact that pseudo code of computing possible discrete values and their corresponding probabilities. However, the process of generating quantile set Q based on I is just one sentence. It would be better if some details of this process is provided in the appendix as well. (Maybe I'm slow in understanding this. I read the code to figure out the process.)

(2) Section 3.3, "We only need to consider the absolute value and the smaller |g|[i] is, the smaller value..." is a little bit confusing. Suggestion: "We only need to consider the absolute value. The smaller |g|[i] is, the smaller value..."

(3) What is "the sign of uniform perturbation"? It's not mentioned previously. Suddenly, the technique is used to deal with catastrophic overfitting.

Please address and clarify the cons and detailed comments above.

**Summary Of The Paper:**

Summary:

The paper introduces a new adversarial training method to improve the performance of Fast adversarial training (FAT), an adversarial training method that depends on only one gradient query. Adversarial training using PGD attack is not efficient, but FAT performs worse than PGD adversarial training. To improve the performance of FAT and maintain the efficiency, the authors study the characteristics of PGD perturbation and proposes a method to imitate PGD perturbation based on only one gradient query.



**Summary Of The Review:**

Reasons for score:


Overall, I vote for weakly accepting. I like the idea of imitating the PGD perturbation to improve the performance of one-step adversarial training. Experiments show that the performance of the proposed method is better than fast adversarial training and has similar training
time. My major concern is that the proposed method is not evaluated on large image dataset, like ImageNet. Some methods perform well
on small datasets like CIFAR10 but do not perform well on large datasets. Since Fast adversarial training can be deployed on ImageNet
(experiments can be found in the paper), why not compare I-PGD-AT with FAT on ImageNet?

##########################################################################
Updates: Thanks for the authors' response. Part of my concerns are addressed by the author. The concern about lack experiments on ImageNet is hard to address during rebuttal period. Therefore, I'll keep my rating.

---

> ### Author Response · Authors · 2021-11-23
> **Response to Reviewer AAHj**
>
> We sincerely appreciate your positive remarks on the problem we try to solve, the intuition, idea, and experimental results of our paper that greatly encourage us, and the valuable suggestions that help improve the quality of our paper. In particular, we thank your time and carefulness on checking both our paper and code. We address your comments as follows.
>
> > Q1: Evaluations on ImageNet.
>
> One of the advantages of Fast-AT is its applicability on ImageNet. Since our I-PGD-AT does not introduce extra computational cost, it could be applicable on ImageNet. We are running the results on ImageNet as you suggested. However, as you pointed out, due to the limited computational resource, we have not obtained the results on ImageNet. But we would continually run it on ImageNet and report the results in the final version. Thanks very much for your understanding.
>
> > Q2: The process of generating quantile set Q based on I is just one sentence. It would be better if some details of this process is provided in the appendix as well.
>
> The calculation for quantile might be hard to follow. To help understand it and the entire algorithm, we provide a simple example of imitating PGD2 with the probability $p=2/3$ as follows.
>
> First, we could obtain the probability $P(|\delta'|[i]=0)=1-p=1/3$, $P(|\delta'|[i]=2)=p=2/3$ and calculate the imitative value and accumulated probability $\mathcal{I}=\\{0:1/3, 2:1\\}$. Suppose we obtain the gradient matrix $g=\left[\begin{array}{ccc}
> 0.2 & -0.3 & -0.1  \\\\
> 0.5 & -0.6 & 0.3 \\\\
> 0.7 & 0.1 & 0.9\\\\
> \end{array}\right]$ at line 5 of Algorithm 1, we need to calculate the start and end value for the range with the portion from $0$ to $1/3$ and from $1/3$ to $1$ on the sorted array $[0.1, 0.1, 0.2, 0.3, 0.3, 0.5, 0.6, 0.7, 0.9]$ (the absolute values of $g$), respectively. Thus, we could get the quantile $Q=\{0.1,0.2,0.9\}$ at Line 6 of Algorithm 1 (0 is the minimal value for the portion of 0, 0.2 is the smaller 3-quantile for the portion of 1/3 and 0.9 is the maximum value for the portion of 1.). After the for-loop at line 7-9 of Algorithm 1, we have $\tilde{g}=\left[\begin{array}{ccc}
>      0 & 2 & 0  \\\\
>      2 & 2 & 2 \\\\
>      2 & 0 & 2\\ \\
> \end{array}\right]$ by calculating each element $\tilde{g}[i] =\left\\{ \begin{array}{cl}
>     0 & \mathrm{if } \ 0.1 \leq |g|[i] \leq 0.2  \\\\
>     2 & \mathrm{if } \  0.2 < |g|[i] \leq 0.9 \\\\
> \end{array}\right.$. Then we let $\tilde{g}$ share the same sign of $g$ at line 10 and obtain $\tilde{g}=\left[\begin{array}{ccc}
>      0 & -2 & 0  \\\\
>      2 & -2 & 2 \\\\
>      2 & 0 & 2\\ \\
> \end{array}\right]$, which could be used to generate the perturbation for adversarial training at line 11-12 of Algorithm 1.
>
> We have added a simple example in detail in Appendix A.2. Thanks very much for your suggestion.
>
>
> > Q3: Section 3.3, "We only need to consider the absolute value and the smaller |g|[i] is, the smaller value..." is a little bit confusing. Suggestion: "We only need to consider the absolute value. The smaller |g|[i] is, the smaller value..."
>
> We have followed your suggestion to clarify this sentence in the revision. Thanks very much for the valuable suggestion.
>
> > Q4: What is "the sign of uniform perturbation"? It's not mentioned previously. Suddenly, the technique is used to deal with catastrophic overfitting.
>
> The sign of uniform perturbation means that we only utilize the sign of the uniform perturbation without considering the value for initialization, i.e., the element of perturbation for initialization is uniformly sampled from $\\{-1, +1\\}$. It is a different initialization in which we want to show that our hypothesis is not constrained on the uniform perturbation for initialization. Since the sign of uniform perturbation might be confused, we change it into the perturbation sampled from $\\{+1, -1\\}$ uniformly for clarity, denoted as FAST-AT-S. We have clarified it in the revision. Thanks very much for pointing it out.

---

### Official Review · Reviewer_Dz2K · 2021-11-01

**Correctness:** 4
**Technical Novelty And Significance:** 2
**Empirical Novelty And Significance:** 2
**Recommendation:** 5
**Confidence:** 5

**Main Review:**

##########################################################################

Pros:

1. The paper focuses on the catastrophic overfitting problem caused by the limited number of adversarial perturbation patterns generated by the single-step attack method during the adversarial training process, and proposes I-PGD-AT to alleviate the problem.

2. The paper pays attention to the both problems of high computations in PGD-based adversarial training and limited attack patterns in FGSM-based adversarial training, and gives a compromise method to absorb the advantages of both methods and discard the disadvantages at the same time.

3. This paper provides comprehensive experiments on CIFAR10 and Tiny ImageNet to demonstrate the effectiveness of I-PGD-AT. And further analysis on catastrophic overfitting problem is provided.

##########################################################################

Cons:

1. The author only focuses on $\delta’$ and ignores $\rho$ in Fig. 1 to demonstrate that PGDk can generate more diverse search space and further find more diverse adversarial examples in the process of attack. However, since both FGSM-RS and PGDk utilizes the random start in the beginning of the attack, I think both of the methods can explores the whole search space in \Deta.

2. Compared to Fast-AT, I-PGD-AT only slightly improves the robustness of the model under attack (average ~2% in Tab. 1), but the classification accuracy on clean images has dropped obviously (the drop also reaches ~2% in Tab. 1).

3. The author has too few methods to compare in the experiments. Since a large number of methods in adversarial training haves been proposed in recent years, more methods should be compared to demonstrate the effectiveness of the I-PGD-AT.

#########################################################################

Some typos:

(1) two lines above Sec. 3.4: \mathcal{I}=\{0: 1-p, 2: 1\} -> \mathcal{I}=\{0: 1-p, 2: p\}

(2) the same problem with (1) in Appendix A.1

(3) immediately after the problem of (2), “arbitra ry” should have no blank space





**Summary Of The Paper:**

The paper notices that the existing methods of adversarial training has the problem of catastrophic overfitting, and proposes an efficient single-step adversarial training method I-PGD-AT by adopting I-PGD attack for training, in which I-PGD imitates the perturbation of PGD based on the magnitude of gradient. Extensive empirical evaluations on CIFAR-10 and Tiny ImageNet demonstrate that I-PGD-AT can improve the robustness compared with the baselines and delay catastrophic overfitting.

**Summary Of The Review:**

Overall, I vote for weak reject. The paper proposes I-PGD-AT to imitate PGD virtually through single gradient calculation. My major concern is about the margin improvement and the limited compared methods in experiments (see cons below). Hopefully the author can address my concern in the rebuttal period.

-------------------------------------------------------------------------------------------------------------------------------------------------------------------------------------

UPDATE

After reading all the response and comments from other reviewers, I will keep my scores. My main concern is the insufficient experiments.

---

> ### Author Response · Authors · 2021-11-23
> **Response to Reviewer Dz2K**
>
> > Q1: Since both FGSM-RS and PGDk utilize the random start in the beginning of the attack, I think both of the methods can explore the whole search space in $\delta$.
>
> As we pointed out in Sec. 3.3, the random start in FGSM-RS indeed helps generate more diverse perturbation, which helps explore the whole search space. That might be the reason why Fast-AT works better than FGSM-AT. However, it is not enough to explore the inner space only through the random start in practice. That would be the reason why PGD-AT works better than Fast-AT since PGD also effectively explores the search space as shown in Fig. 1. Thus, we propose I-PGD-AT to boost Fast-AT by imitating the *diverse* perturbation generated by PGD without extra cost. The performance improvement also supports the effectiveness of our method.
>
> > Q2: Compared to Fast-AT, I-PGD-AT only slightly improves the robustness of the model under attack (average ~2% in Tab. 1), but the classification accuracy on clean images has dropped obviously (the drop also reaches ~2% in Tab. 1).
>
> This paper aims to improve the robustness of Fast-AT **without extra computational cost** to maintain the high training efficiency. With the simple imitation strategy, the improvement on Fast-AT is even better than GradAlign, which strengthens the local linearity through double backpropagation and leads to $3\times$ computational cost. Thus, the improvement is indeed significant considering the rigorous restriction of no extra computational cost.
>
> As for the classification accuracy on clean images, there is usually a tradeoff between benign accuracy and robust accuracy. Kim et al achieves better benign accuracy but also degrades the robust accuracy while GradAlign exhibits better robust accuracy but lower benign accuracy than Fast-AT. A similar tradeoff could also be observed in PGD-AT as shown in Table 2. With the slightly better benign accuracy than GradAlign and the significant improvement on the robust accuracy, we think the drop of classification accuracy on clean images is acceptable. This work also provides a new perspective to bridge the gap between PGD-AT and Fast-AT without extra computational cost, which would shed light on further improving Fast-AT while maintaining the high training efficiency.
>
> > Q3: The author has too few methods to compare in the experiments.
>
> In this work, we mainly focus on improving Fast-AT (single-step adversarial training) **without any extra computational cost**, which are rarely investigated. Most single-step adversarial training methods aim to improve the robustness or stabilize the training process but also introduce an extra computational cost. Thus, we only choose these methods as our baselines in the submitted version. Following your suggestion, we also add Free-AT [1] and FastAdv+ [2] as our baselines on CIFAR-10 in Table 1, which further validates the effectiveness of our proposed method.
>
> > Q4: Some typos.
>
> Thanks for pointing out the typos. We have corrected the typos and polished the paper again. It is also noted that $\mathcal{I}$ records the accumulated probability for calculating the quantile instead of the probability for each candidate value. We have clarified it in the appendix and $\mathcal{I}=\\{0: 1-p, 2: 1\\}$ is correct in the paper.
>
> [1] Shafahi et al. Adversarial Training for Free! NeurIPS 2019.
>
> [2] Li et al. Towards understanding fast adversarial training. arXiv preprint arXiv:2006.03089 2020.

---

### Official Review · Reviewer_bLbt · 2021-11-03

**Correctness:** 3
**Technical Novelty And Significance:** 2
**Empirical Novelty And Significance:** 2
**Recommendation:** 3
**Confidence:** 5

**Main Review:**

1 . I wonder if the approximation in (6) makes sense? In (6), the gradient in later iterations can go far outside the perturbation limit and thus make the update nowhere near the actual case where all PGD iterations’ gradients are strictly limited within the perturbation ball.

2 . The authors claimed that one of the problems of one-step AT is that the perturbations are not diverse enough while in Figure 1, they showed that PGD-k should have diverse discrete perturbations. However, in practice, PGD attacks are usually located on the boundary with common choice attack step size (also mentioned in [1]).

[1] "Parsimonious black-box adversarial attacks via efficient combinatorial optimization." International Conference on Machine Learning. PMLR, 2019.

Also, even if Figure 1 is correct or aligned with practice, to me, it still does not explain how adopting random initialization can mitigate such a phenomenon.

3 . In the experiments, I notice that the authors actually used a large attack step size such as epsilon or 1.25epsilon, so that the final projection point is no longer located at those discrete points as in Figure 1. Rather, it will again be located on the boundary of the norm ball.

4 . The robustness improvement is marginal compared with Fast-AT as shown in Table 1. Also, it seems that the proposed method is actually trading natural accuracy for robustness?

5 . In table 2, PGD2-AT performance is even worse than Fast-AT which is clearly not reasonable. The authors might want to double-check their experimental results.

6 . From figure 3, it seems that both I-PGD-AT and Fast-AT are experiencing catastrophic overfitting problems, just that Fast-AT happened first. Since this phenomenon may occur with randomness, it is hard to decide whether I-PGD-AT really solves the catastrophic overfitting problem. The authors should at present the result for repeated runs and see whether this is consistent.

7 . The following two papers also work on fast adversarial training. The authors might also want to comment and compare with them:

[2] "Towards understanding fast adversarial training." arXiv preprint arXiv:2006.03089 (2020).
[3] "Efficient robust training via backward smoothing." arXiv preprint arXiv:2010.01278 (2020).





**Summary Of The Paper:**

In this paper, the authors studied improving the efficiency of adversarial training. The authors first analyzed the difference of perturbation generated by FGSM-RS and PGD and proposed an efficient single-step adversarial training method I-PGD-AT by adopting I-PGD attack for training. Extensive empirical evaluations on CIFAR-10 and Tiny ImageNet demonstrate that the proposed I-PGD-AT can improve the robustness compared with the baselines and significantly delay catastrophic overfitting.

**Summary Of The Review:**

The intuition of the proposed method is not really making sense and the empirical results are not quite convincing, therefore, I recommend reject for this paper

---

> ### Author Response · Authors · 2021-11-23
> **Response to Reviewer bLbt (1/2)**
>
> > Q1: Does the approximation in Eq. (6) make sense?
>
> Our motivation and novelty are to imitate the perturbation generated by iterative PGD attack with single gradient calculation to improve the effectiveness of adversarial training. Thus, in Eq. (6), we recursively expand the equation for PGD attack and approximately omit the projection operator at each iteration to get the final sum, which is rough but motivates us to analyze and utilize the sign sequence to imitate the iterative PGD. The primary purpose of Eq. (6) is to describe our motivation that we could imitate the perturbation generated by iterative PGD with a single gradient calculation rather than to drive a precise approximation of PGD attack. And the empirical evaluation validates the effectiveness of our method.
>
> Regarding your concern that the gradient in later iterations can go far outside the perturbation limit, we first address it without considering the random initialization for simplicity and using the widely accepted setting $\alpha=\epsilon/4=2/255$. Before the adversarial example reaches the boundary, there is no difference between the approximation and PGD. When it reaches the boundary at t-th iteration and exceeds the boundary at the next iteration, the adversarial example would stay at the boundary in PGD until the attack finishes due to the projection. In contrast, in the approximation, it might go far away from the boundary with the iteration progressing on. However, it does not matter because we only have two directions +1 and -1 (the sign of gradient). It cannot change the direction of perturbation outside the boundary and would obtain the same data point on the boundary after the projection when the attack finishes. **Hence, the imprecise gradient in the later iterations does not influence the final results.** With the random initialization, PGD might exceed the boundary in which the data point is not on the boundary before crossing over the boundary, but the distance from this data point to boundary would be smaller than $\alpha=2/255$. With such a smaller distance and the local stability of deep models, the sign of gradient would be consistent with high probability, and it would not make much difference.
>
> > Q2: PGD attacks are usually located on the boundary with common choice attack step size. Even if Figure 1 is correct or aligned with practice, to me, it still does not explain how adopting random initialization can mitigate such a phenomenon.
>
> **The perturbation on the boundary does not make a difference for PGD and FGSM-RS and we should focus on the perturbation in the boundary.**
>
> As shown in Fig. 1 in [1], around 90% of the perturbation is on the boundary, which does not make any difference because a single-step attack with a large step size would lead to the same results on the boundary. As shown in Fig. 2 in our paper, the sign consistency is very high ($\geq$90%), which also supports that most elements in the perturbation of PGD would be at the decision boundary. And we should focus on the remaining 10% of the perturbation in the boundary, highlighting the difference between the perturbation generated by FGSM-RS and PGD.  This also makes it easier for us to imitate the perturbation generated by PGD with a single gradient calculation.
>
> **The random initialization diversifies the perturbation.**
>
> Without the random initialization, the perturbation generated by FGSM would entirely be at the boundary. With the random initialization, the elements with different signs of gradient and signs of the random initialization would be in the boundary, diversifying the generated perturbation as in PGD. Hence, the random initialization would mitigate the catastrophic overfitting. We have made it more clear in the revision.
>
> > Q3: In the experiments, I notice that the authors actually used a large attack step size such as epsilon or 1.25epsilon, so that the final projection point is no longer located at those discrete points as in Figure 1. Rather, it will again be located on the boundary of the norm ball.
>
> In our experiments, we adopt the attack stepsize with $\epsilon$ due to single step gradient calculation. However, note that we adopt the imitated gradient instead of the sign of gradient to craft the perturbation, which would still lead to the perturbation within the boundary. For instance, there is 0 in our imitated gradient for I-PGD2-AT, which would make the corresponding perturbation in the boundary.
>
> [1] Moon et al. Parsimonious black-box adversarial attacks via efficient combinatorial optimization. ICML 2019.

---

> > ### Author Response · Authors · 2021-11-23
> > **Response to Reviewer bLbt (2/2)**
> >
> > > Q4: The robustness improvement is marginal compared with Fast-AT as shown in Table 1. Also, it seems that the proposed method is actually trading natural accuracy for robustness?
> >
> > In general, our I-PGD3-AT improves the performance of roughly 2% on CIFAR-10 for Fast-AT **without extra computational cost** using such simple imitation. In particular, our I-PGD-AT exhibits better results than GradAlign, which takes $3\times$ computational cost due to the double backpropagation. On TinyImagenet, since the high and stable sign consistency, the improvement is not so remarkable but still larger than 0.6%, showing the effectiveness of our imitation.
> >
> > There is a tradeoff between natural accuracy and robust accuracy, which has been validated by many works. For instance, as shown in Table 1, GradAlign exhibits higher robust accuracy but lower natural accuracy than Fast-AT, while kim et al. exhibits higher natural accuracy and lower robust accuracy than Fast-AT. Also, when we adopt larger iterations for PGD-AT, the natural accuracy decreases while the robust accuracy increases. Our main contribution is to provide a way to imitate the iterative perturbation generated by PGD with a single gradient calculation to train a robust model efficiently. Such decay on natural accuracy is acceptable considering the improvement on robustness.
> >
> > > Q5: In table 2, PGD2-AT performance is even worse than Fast-AT which is clearly not reasonable. The authors might want to double-check their experimental results.
> >
> > Our code is adopted from [1]. We have double-checked the experiment setting and code to make sure the results are reliable. Fast-AT follows the training setting in [2] with the cycle learning rates, while PGD2-AT is trained with piecewise learning rate without early stopping as in [3]. We find that cycle learning rate plays a crucial role in improving the performance of Fast-AT, which explains why PGD2-AT performance is even worse than Fast-AT in Table 2. Also, we find that the robust accuracy of PGD2-AT on PreAct ResNet-18 against AA is $34.8%\pm 2.1$ in Table 1 of [4], which is similar to our result (36.52%) in Table 2 under the similar setting, validating that the results are trustworthy.
> >
> > > Q6: Catastrophic overfitting for I-PGD-AT.
> >
> > As discussed in Sec. 4.5, I-PGD-AT could delay the catastrophic overfitting but cannot avoid catastrophic overfitting. We have run the experiments several times and found I-PGD-AT does not lead to catastrophic overfitting in the first 30 epochs and exhibits consistency on the robustness.
> >
> > > Q7: The authors might also want to comment and compare with [5] and [6].
> >
> > This paper aims to improve the vanilla Fast-AT **without extra computational cost**. Thus, we chose three single-step adversarial training methods as our baselines. Following your suggestion, we have discussed these two works in the related work.
> >
> > FastAdv+ [5] adopts PGD-AT when observing catastrophic overfitting in Fast-AT to stabilize the training, which introduces extra training costs due to using the iterative PGD. We also take it as our baseline on CIFAR-10. As shown in Table 1, our I-PGD-AT exhibits slightly better robustness than FastAdv+ with a much lower training cost, demonstrating the effectiveness of I-PGD-AT.
> >
> > Chen et al. [6] propose *backward smoothing*, which initilizes the perturbation with backpropagation to accelerate the training of TRADES and we have discussed it in the related work. As mentioned in Sec. 5.4 [6], *backward smoothing* exhibits very good performance on TRADES but could not bring significant performance improvement for Fast-AT, using two backpropagations for generating adversarial examples. Since we mainly focus on Fast-AT, which adopts the loss function of Cross-Entropy, we do not add it as our baseline.
> >
> > [1] Croce et al. Reliable Evaluation of Adversarial Robustness with an Ensemble of Diverse Parameter-free Attacks, ICML 2020.
> >
> > [2] Eric et al. Fast is better than free: Revisiting adversarial training. ICLR 2020.
> >
> > [3] Madry et al. Towards Deep Learning Models Resistant to Adversarial Attacks. ICLR 2018.
> >
> > [4] Kim et al. Understanding Catastrophic Overfitting in Single-step Adversarial Training. AAAI 2021.
> >
> > [5] Li et al. Towards understanding fast adversarial training. arXiv preprint arXiv:2006.03089 2020.
> >
> > [6] Chen et al. Efficient robust training via backward smoothing. arXiv preprint arXiv:2010.01278 2020.

---

> > > ### Comment · Reviewer_bLbt · 2021-11-30
> > > **Further Comments**
> > >
> > > I thank the authors for their detailed response. However, the response still does not address some of my main concerns about the paper.
> > >
> > > For Q1, the authors admit that "in the approximation, it might go far away from the boundary with the iteration progressing on". Then the authors argue that this will not change the direction of perturbation outside the boundary and would obtain the same data point when the attack finishes, which I don't believe is, in any sense, correct. The gradient at different positions could be drastically different even in sign directions.
> > >
> > > For Q2, the authors' claim is also contradictory. They claim that the goal is to imitate the behavior of PGD. Yet PGD leads to 90% perturbation on the boundary while the authors say they now focus on the remaining 10% perturbations within the boundary. If so, it is not imitating the behavior of PGD. Also, there is no rigorous relationship between diversifying the perturbation and catastrophic overfitting in the current literature. As shown in the GradAlign paper, FastAT with a smaller $\epsilon$ without random init can still work but it does not diversify the perturbation. The current hand-wavy argument is not really convincing.
> > >
> > > For Q3, the perturbation is on the Linf norm boundary as long as one of the dimensions reaches $\epsilon$. Therefore, even if there exists 0, it seems to still have an extremely high probability to stay on the boundary with such a large step size?
> > >
> > > Based on the above points, I would keep my original opinion and score.

---

### Decision · Program_Chairs · 2022-01-20

**Decision:**

Reject

**Comment:**

This paper aims to improve the efficiency of adversarial training. Specifically, by analyzing the differences between the adversarial perturbations generated by FGSM-RS and the adversarial perturbations generated by PGD, this paper proposes a new single-step attacker I-PGD (which imitates PGD by creating diverse adversarial perturbations) to accelerate adversarial training. Empirical results are provided on CIFAR-10 and Tiny ImageNet to support the effectiveness of the proposed method.

Overall, the reviewers think it is an interesting paper, but are severely concerned about some statements. During the discussion period, the authors actively clarify these points. However, the Reviewer bLbt is not fully convinced and believes 1) the approximation in Eq. (6) is incorrect and 2) the proposed method is loosely motivated by imitating the behavior of PGD. The authors fail to further follow up on this discussion. The Reviewer bLbt and the Reviewer Dz2K are also concerned that the proposed I-PGD-AT only yields margin improvements over Fast Adversarial Training. In addition, as suggested by the reviewer AAHj, given this work focuses on developing efficient adversarial training, it is important to include results on larger-scale datasets like ImageNet.

I encourage the authors to incorporate all the reviewers' comments and make a stronger submission next time.